# Microstructure and Corrosion Behavior of Fe-Based Austenite-Containing Composite Coatings Using Supersonic Plasma Spraying

Xiaoyan Zhang [1,2], Tiegang Luo [1,*], Shenglin Liu [2,*], Zhibin Zheng [1], Juan Wang [1], Kaihong Zheng [1], Shuai Wang [1] and Huantao Chen [1]

1   Institute of New Materials, Guangdong Academy of Sciences, National Engineering Research Center of Powder Metallurgy of Titanium& Rare Metals, Guangdong Provincial Key Laboratory of Metal Toughening Technology and Application, Guangzhou 510650, China; 17809162428@163.com (X.Z.)
2   School of Materials Science and Engineering, Chang'an University, Xi'an 710061, China
*   Correspondence: luotiegang2021@163.com (T.L.); 17691263671@163.com (S.L.)

**Abstract:** The Fe-based austenite-containing composite coatings with various contents (3 vol.%, 6 vol.%, 9 vol.%, 12 vol.%) of austenite powder additions were created by supersonic plasma spraying on 45 steel substrates. The microstructure, phase composition, microhardness, and porosity of the composite coatings were examed. Moreover, special attention was paid to the effect of austenite powder on the corrosion resistance of the austenite-containing composite coatings. The results found that the addition of austenite powders could significantly improve the corrosion resistance of Fe-based coatings, which is mainly due to three correlated phenomena caused by the austenite particles. First, austenite particles significantly reduce the porosity of the austenite-containing composite coatings and form a denser coating structure due to their low melting point and good chemical compatibility with the Fe-based alloy. Further, austenite particles help to refine the grains and increase the grain boundary density. Last but not least, austenite particles help to generate more diffusely distributed second phases in the coating, improving the chemical homogeneity and hardness of the coating.

**Keywords:** Fe-based composite coating; supersonic plasma spraying; corrosion resistance; austenite powders





## 1. Introduction

One of the primary reasons for the material failure of metal equipment is corrosion, which seriously affects the service life and service reliability of equipment and causes huge economic losses [1–5]. Using coatings to enhance the surface characteristics of materials is an efficient technique to lengthen the service life of components. Because of its advantages of ultra-high hardness, excellent wear resistance, and high-cost performance, Fe-based self-melting powder has shown great commercial value in the field of thermal spraying [6–12]. However, despite the popularity of iron-based coatings due to their superior comprehensive properties, such materials have the fatal disadvantage of poor corrosion resistance, which greatly limits their application [13,14]. Expanding the application's reach and lowering production costs are thus extremely important.

It is well known that the content and structure of coatings have a significant impact on their ability to resist corrosion [15–17]. By quickly producing an oxide film in a corrosive environment, corrosion-resistant elements like Cr, Si, and Ni can increase the corrosion resistance of coatings. It is worthwhile to try to alter the coating's corrosion resistance by modifying the content of corrosion-resistant elements from the standpoint of the composition design of thermal spray coating. There are not many publications on incorporating austenite powder into the Fe-based coating's microstructure as of yet.

Qiang Wang et al. studied the wear action of the Fe-based coating prepared using supersonic plasma spraying. The study demonstrated that the wear mechanism of the coating contained the two mechanisms of oxidation and fatigue wear, while this work did not analyze the coating's resistance to abrasion [18]. Li Fan et al. studied the Fe-based coating made using laser fusion in 0.5 mol/L HCl solution for its resistance to electrochemical corrosion. The high-density oxidation film's mechanical barrier was determined to be responsible for the coating's superior corrosion resistance, according to the results [19]. Cong Li et al. thoroughly studied how chromium affected the interfacial bonding and erosion wear resistance of a multi-arc ion plating prepared high chromium white cast iron (ZPAT/Fe) composite coating. The outcomes demonstrated that adding chromium might undoubtedly increase abrasion resistance. The composite with additional chromium had erosion wear resistance that was 1.44 times greater than the composite without chromium [20]. The effects of $B_4C$, MoB, and TaB on the structure, phases, microhardness, and wear rate of Fe-based self-fusing alloy coatings were investigated by E et al. The addition of the particles increased the hardness as well as the wear resistance of the coatings. The best results were obtained in the presence of TaB particles: up to 2 times for single alloying and up to 1.6 times for multi-component alloying [10].

So far, especially in the simultaneous elucidation of the behavior and performance of Fe-based coating in corrosion and erosion environments, there have not been many research works on the corrosion resistance of thermal spraying Fe-based alloy coating [21–23]. In addition, from the perspective of composition design, the corrosion and erosion behavior of coating, that the austenitic stainless steel powder was added to the Fe-based coating using supersonic plasma spraying technology, and the corrosion resistance of the Fe-based composite coating was enhanced by changing the content of corrosion resistance elements. This has not been studied.

Supersonic plasma spraying is a novel type of thermal spraying technology successfully created in the 1990s. It has the advantages of fast particle flight speed, high deposition efficiency, and good coating quality. High material fusing temperatures and particle jet velocities are two characteristics that make the plasma spraying method stand out. In this research background, the austenite powder-reinforced Fe-based composite coating was produced using supersonic plasma spraying (SPS) technology, and the microstructure and corrosion resistance of the coating were investigated.

## 2. Materials and Methods

### 2.1. Coating Preparation

For the raw materials, we used commercially available Fe60 alloy powder (Nangong Xindun Alloy Welding Spray Co., Ltd., Xingtai, China) and austenite powder (Hebei Yangyou Metal Materials Co., Ltd., Baoding, China) with regular sphericity, as shown in Figure 1. The powder's average particle size was 65 and 33 μm, respectively. Table 1 displays the chemical makeup of the Fe60 self-fluxing metal particles.

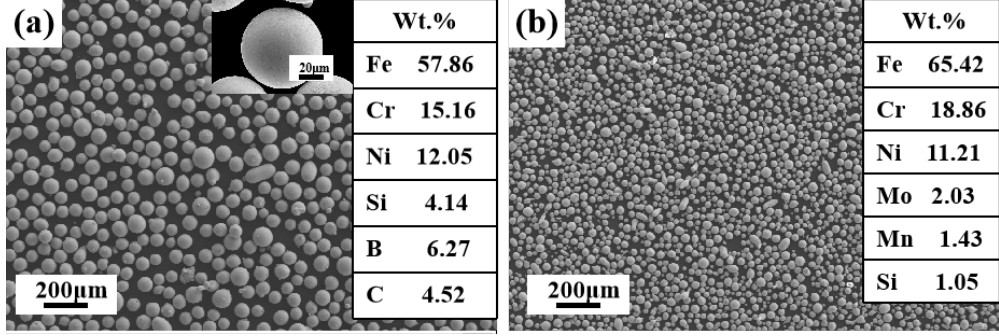

**Figure 1.** Morphology of (**a**) Fe60 and (**b**) austenite powder.

**Table 1.** Composition of the Fe60 powder.

| Elements | C | Si | B | Cr | Ni | Fe |
|----------|-----|-----|-----|------|------|------|
| Fe60 | 0.8–1.2 | 1.0–2.0 | 3.8–4.2 | 16–18 | 9.0–12 | Bal. |

Low-energy ball milling was used to mix Fe60 powder with 3 vol.%, 6 vol.%, 9 vol.%, and 12 vol.% austenite powder. We set the speed of the three-dimensional mixing instrument equipment to 30 rpm and the mixing time to 12 h, then get a uniformly mixed Fe-based composite powder. A bulk 45 steel substance, measuring 100 mm × 100 mm × 30 mm, served as the experiment's substrate. Before spraying, the substrates should be cleansed with an absolute alcohol solution to remove greasy grime, followed by an alumina abrasive sandblasting to create a clear surface and strengthen the binding between the substrate and coating even more. The spraying was carried out using the supersonic plasma spraying system [24,25]. The alloy powder was dried at 393 K to eliminate any remaining water in order to maximize the fluidity of the powder. Compressed air was used to maintain a low temperature ($\leq$200 °C) during the spraying operation in order to lower the danger of overheating and coating oxidation. Ar was used as the primary gas and the carrier gas, while $H_2$ served as the backup gas. Table 2 displays the detailed spraying parameters of supersonic plasma spraying. The coatings produced have a thickness of around 350 microns. To prevent confusion, we named the coating prepared from Fe60 powder without the addition of austenitic powder as a Fe-based coating, and named the Fe-based composite coatings prepared by adding austenite powder (3 vol.%, 6 vol.%, 9 vol.%, 12 vol.%) to Fe60 powder as x% $\gamma$-Fe coating (x = 3, 6, 9, 12), respectively.

**Table 2.** The processing parameters of the SPS.

| Power/KW | Ar/(L/min) | $H_2$/(L/min) | Powder Feed Rate (g/min) | Spray Distance (mm) |
|----------|------------|---------------|--------------------------|---------------------|
| 55 | 100 | 6 | 25 | 90 |

### 2.2. Characterisation of the Coating

The microstructure of the coatings was examined using a scanning electron microscope (SEM, ZEISS, Jena, Germany) coupled with an electron backscattered diffractometer (EBSD) detector (EBSD, ZEISS, Jena, Germany). The CHANNEL 5.0 software performed further analysis of the EBSD data. By using X-ray diffraction (XRD) with Cu $K_{\alpha}$ radiation, the phase structure of the coating was examined. By randomly selecting more than 10 cross-sectional SEM images of the coating at 800× magnification, the porosity results were calculated and averaged using porosity assessment software (Image J2). Using an HVS-1000 type microhardness tester with a 100 g applied force and a 10 s duration period, microhardness was assessed. Each coating sample was measured at least 8 times, and the average hardness value was calculated after removing the highest and lowest values. The immersion corrosion test specimens were resin-mounted, and the working surface of size 10 × 10 $\mu m^2$ was abraded and polished and then subjected to immersion corrosion tests in 3.5 wt % NaCl solution for 1, 3, and 7 days, respectively. The immersion surfaces were characterized using an SEM after the immersion corrosion test.

### 2.3. Electrochemical Corrosion Test and Immersion Corrosion Test

Platinum was employed as the counter electrode for the electrochemical experiments, and the working electrode was the coating sample that would be examined, while the reference electrode was a saturated calomel electrode, which was conducted using a Gamry electrochemical workstation (INTERFACE1000, Indiana, USA) with a standard three-electrode cell at room temperature in a freely aerated 3.5 wt.% NaCl solution. The exposure area of the coating sample was 1 $cm^2$. Furthermore, the sample was polished with 200#–2000# sand paper and polished to mirror before testing. Finally, the sample was ultrasonically

cleaned with anhydrous ethanol for 20 min. The observation time of the open circuit potential is about 3600 s before starting the potentiodynamic polarization test. At a scan rate of 0.5 mV/s, the potentiodynamic polarization experiments were conducted. To guarantee the results' repeatability, three parallel specimens were employed. The obtained potentiodynamic polarization data were analyzed using CorrTest software (Version 4).

*2.4. Erosion–Corrosion Test*

A liquid-solid two-phase flow erosion wear tester (MF-20,Hansen, Shanghai, China) was used to perform the erosion–corrosion test. The corrosive medium was a 3.5 wt.% sodium chloride solution containing white corundum particles (46#) with a volume ratio of 5:4. The sample was immersed in an erosion medium during the test. A 20 m/s speed was used for the weight loss test, an erosion–corrosion Angle of 60°, and an exposed area of 360 mm$^2$. Before testing, the surface of the coating was first sanded to a smooth surface and then washed in alcohol, dried, and weighed at the end. To guarantee that the data were reliable, each sample was tested at least three times in total. After the erosion–corrosion test, SEM was used to investigate the erosion morphology. The roughness of the surface subjected to the erosion–corrosion test was determined using atomic force microscopy (AFM; ParkSystems, Korea), and the three-dimensional morphology was observed.

## 3. Results and Discussion
### 3.1. Microstructure and Porosity of Coatings

Figure 2 shows the cross-sectional SEM morphologies of Fe-based coating and austenite-reinforced Fe-based composite coatings (taking 6% γ-Fe coating and 9% γ-Fe coating as an example). All coatings had a similar thickness of about 350 μm. In the high-magnification picture in Figure 2d–f, it can be observed that pores, oxides, and unmelted particles existed in the coating. It is noteworthy that from the EDS results, the appearance of oxide layers in the coating is similar to other works [7,26]. It can be observed that, as the austenite content increased, the unmelted particles in the coatings became fewer and fewer, and as the austenite powder was fully melted during the spraying process, the pores in the Fe-based composite coatings were filled, resulting in a decreasing number of pores in the coatings. Therefore, it can be concluded that Fe-based composite coatings have a higher corrosion resistance than the Fe-based coating because of lower porosity and adhesive structure [27–29]. Moreover, as seen in Figure 2d–f, a thin layer of black strip forms on the interface. The SiO$_2$ phase forms on the interface and is pinned into the interface, according to the EDS of the coatings, which increases the binding capacity of the substrate and coating [30].

Figure 3 shows the Inverse pole figure (IPF) results for the coatings measured using EBSD. In the grain boundaries maps Figure 3d–f, the blue line characterizes the low-angle grain boundaries (LAGBs) ($\theta \leq 15°$), and the black line characterizes the high-angle grain boundaries (HAGBs) ($\theta > 15°$). The volume percentage of HAGBs is clearly larger than that of LAGBs in all coatings, and because recrystallization will always occur, only a small amount of LAGBs will be produced. Moreover, according to Figure 3g–i, with the increasing content of austenite, the amount of LAGBs increases. It has been demonstrated that LAGBs can significantly reduce the onset and progression of corrosion [31,32]. According to OIM Analysis6 software statistics, the grain sizes of 6% γ-Fe coating and 9% γ-Fe coating are 2.3 and 2.1 μm, respectively, which is slightly lower than that of Fe-based coating (2.6 μm). This can be ascribed to the increased Heterogeneous nucleation of carbides at grain boundaries. The inhomogeneous grain size ranges from sub-micrometer to more than 5 μm. Due to the more ultra-refined grains creating more grain boundaries, the 9% γ-Fe coating displays a substantially greater grain boundary density than the Fe-based coating. The bulk of studies has shown that grain size has an impact on corrosion resistance. The primary factor increasing the density of grain boundaries is the much smaller grain size, which leads to a more corrosion-resistant coating [33,34]. Therefore, due to their

smaller grain size and higher density of their grain boundaries, it can be concluded that the Fe-based composite coatings offer improved corrosion resistance.

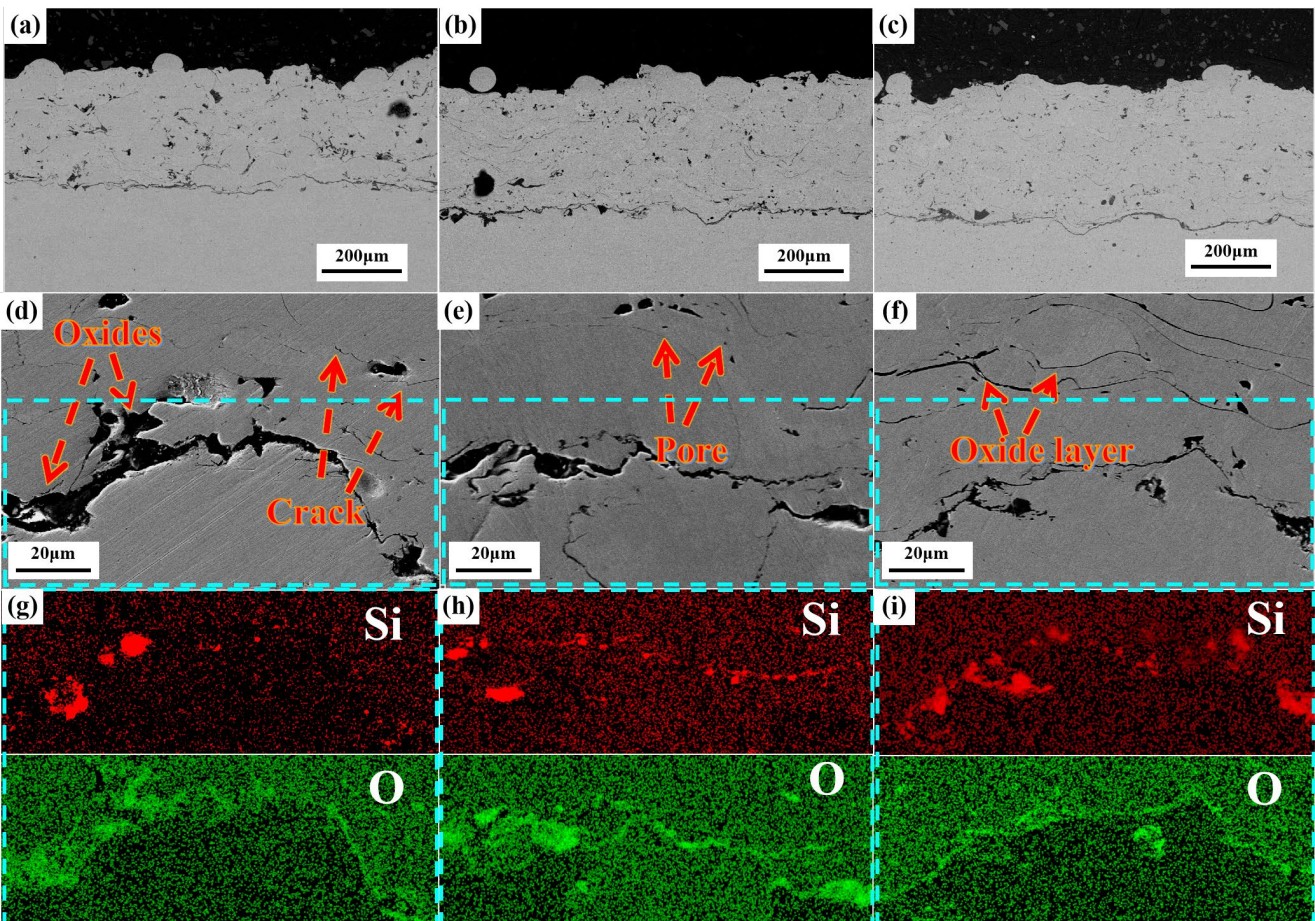

**Figure 2.** Cross-sectional SEM morphologies and EDS analysis of the coatings: (**a**,**d**,**g**) Fe-based coating, (**b**,**e**,**h**) 6% γ-Fe coating, (**c**,**f**,**i**) 9% γ-Fe coating.

The porosity quantity of the coatings is measured in Figure 4. The cross-section morphology of coatings was captured in five SEM pictures at 800× magnification, which was utilized to determine the average porosity value. Based on the picture analysis, the porosity measurement was assessed. In the end, the porosity values for 6% γ-Fe coating and 9% γ-Fe coating were 1.02% and 0.93%, respectively, which is much less than that of Fe-based coating (3.97%).

According to Figure 5, we can see clearly that the austenite particles used in this research were embedded into the Fe substrate very well. The porosity of Fe-based composite coating was significantly improved by the addition of austenite powder. This is mainly owing to the melting of the austenite powder during high-temperature thermal spraying (Figure 5b,c), which increases the binding area between the Fe-based alloy particles and the austenite particles. By tightly combining with the deformed Fe-based alloy particles in the composite coating and filling any gaps or voids left by the latter, austenite particles efficiently raised the density of the coating. With an increase in austenite particle content, the coating porosity dropped. This is because more and more austenite particles were fully melted, and many of the internal voids in the well-flattened second phase disappeared; thus, the average porosity remained at a low level. It is well known that low porosity results in good corrosion resistance of coating [22,35].

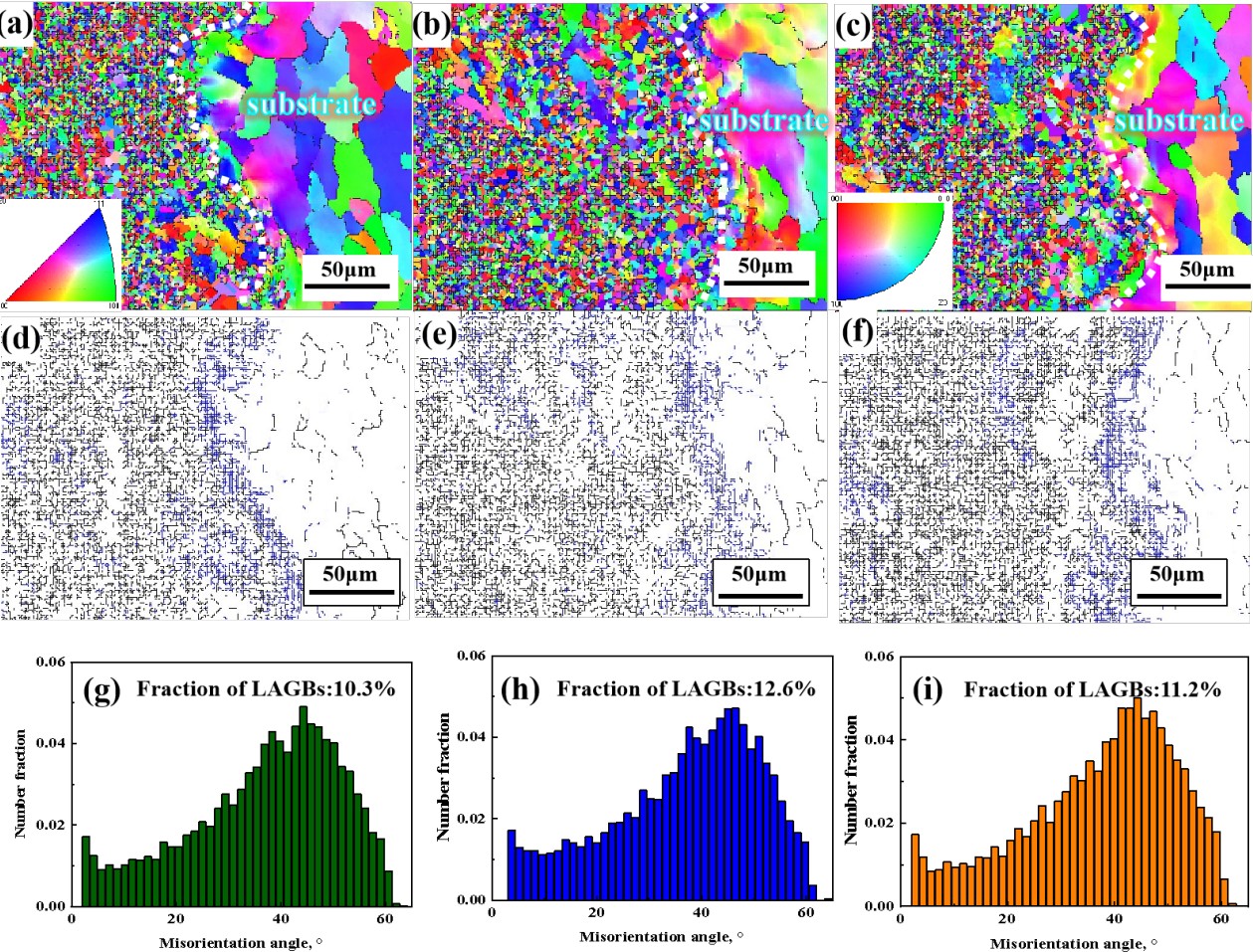

**Figure 3.** IPF maps, distinguishing grain boundary distribution and fraction of LAGBs of the coatings, respectively: (**a**,**d**,**g**) Fe-based coating, (**b**,**e**,**h**) 9% γ-Fe coating, (**c**,**f**,**i**) 9% γ-Fe coating.

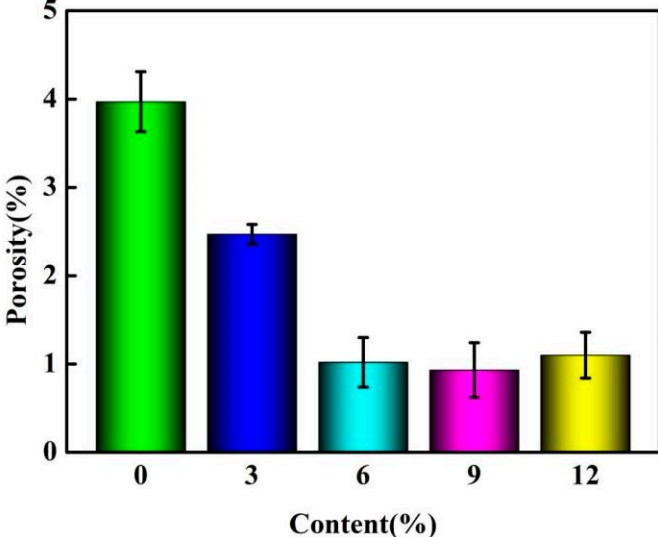

**Figure 4.** The porosity of the coatings.

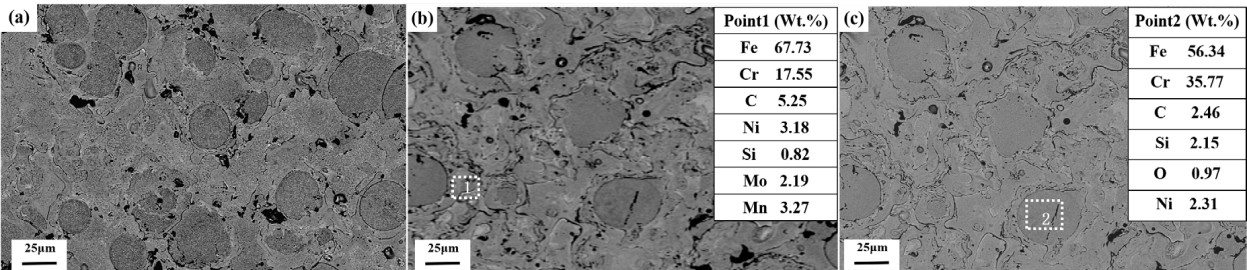

**Figure 5.** Surface SEM morphologies of the coatings: (**a**) Fe-based coating, (**b**) 6% γ-Fe coating, (**c**) 9% γ-Fe coating.

### 3.2. Phase Compositions and Microhardness of Coatings

The XRD pattern of the coatings is shown in Figure 6. It can be seen that α-Fe, $Cr_7C_3$, and $Cr_{23}C_6$ are the main phases in the coating. Once austenite particles were included in the coating, there appeared γ-Fe and $Fe_3C$ phases. The XRD pattern of 6% γ-Fe coating is similar to that of 9% γ-Fe coating.

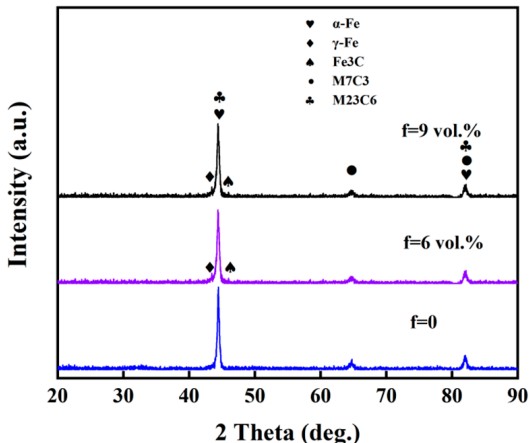

**Figure 6.** XRD diffraction pattern analysis of the coatings.

Figure 7 displays the phase distribution of the coating as determined by EBSD. The phase composition is summarized in Table 3, and the f represents the volume percentage of austenite powder added to the Fe-based alloy powder. The EBSD results are consistent with the XRD. The three main reinforcement phases are $Cr_{23}C_6$, $Cr_7C_3$, and $Fe_3C$, respectively. These carbides in the coating form a reinforced mesh skeleton, as shown in Figure 7a–c. Furthermore, it is clear that in the grains of three coatings, a small amount of the γ-Fe phase has precipitated. With an increase in austenite powder content, the content of the γ-Fe phase rises significantly. According to the report from Mariko Kadowaki et al. [36], retained γ-Fe could act as a barrier against pit propagation which made the Fe-based composite coating have improved resistance to corrosion. Therefore, the presence of the γ-Fe phase enhances the ability of the coating to resist corrosion.

**Table 3.** Phase content of different coatings.

| Sample | BCC | $Cr_{23}C_6$ | $Cr_7C_3$ | $Fe_3C$ | FCC |
|--------|-----|--------------|-----------|---------|-----|
| f = 0 | 49.9 | 23.5 | 1.04 | 1.68 | 0.19 |
| f = 6 vol.% | 47.6 | 28.1 | 1.44 | 0.62 | 0.23 |
| f = 9 vol.% | 46.7 | 31.2 | 1.28 | 0.54 | 0.31 |

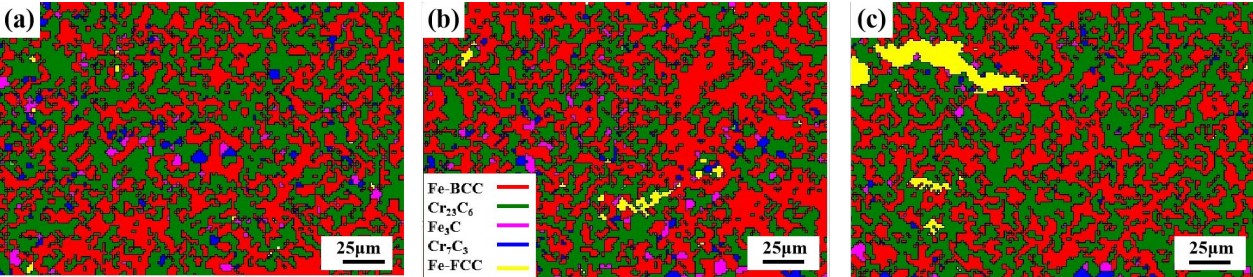

**Figure 7.** Phase distribution of different coatings from EBSD: (**a**) Fe-based coating, (**b**) 6% γ-Fe coating, (**c**) 9% γ-Fe coating.

Figure 8 shows the microhardness of the coating (Figure 8). It is evident that among all coatings, the Fe-based coating exhibits the lowest microhardness, which is around 562HV$_{0.1}$. The microhardness of the Fe-based composite coatings clearly increases as austenite particle content rises. The hardness values for 6% γ-Fe coating and 9% γ-Fe coating are 606HV$_{0.1}$ and 634HV$_{0.1}$, respectively, which are markedly higher than the hardness values of the Fe-based coating. These values are due to the uniform distribution of the added austenite powder in the Fe-based composite coating and the formation of some metallic carbides phases during the thermal spraying process, such as $Cr_{23}C_6$, $Cr_7C_3$, and $Fe_3C$, which diffused into the coatings and significantly increased the dispersion strengthening (Figure 2). In addition, this explains why Fe-based composite coatings have better microhardness than Fe-based coatings [15,37].

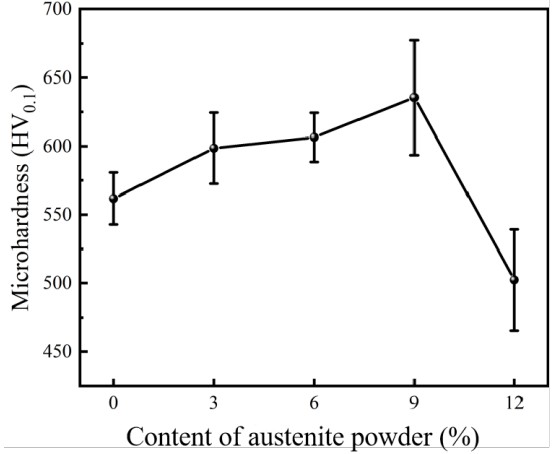

**Figure 8.** Microhardness of the coatings.

Among all samples, the 9% γ-Fe coating has the highest hardness value (634HV$_{0.1}$), which can be mainly ascribed to the following: On the one hand, the addition of 9 vol.% austenite powder causes the precipitation of $Cr_{23}C_6$, $Cr_7C_3$, and $Fe_3C$ phases in addition to raising the relative amounts of various compounds in the Fe-based composite coating result in the second phase strengthening. On the other hand, the addition of austenite powder improves the microstructure of the Fe-based composite coating, leading to fine-grain strengthening. Moreover, the uniform dispersion of thin and fine oxide layers in the coating, which gives the Fe-based composite coating a very effective oxide dispersion strengthening effect, is another factor contributing to the high hardness of 9% γ-Fe coating. When the doping content of austenite particles increased to 12 vol.%, the coating hardness decreased slightly. This is due to the fact that the doped austenite powder itself will also form the pore while filling the pore of the Fe-based alloy powder. Therefore, when the doping amount reaches a critical value, the reduction of the porosity of the austenite powder to the Fe-based alloy powder is not enough to compensate for the porosity brought by itself. At this time, the porosity increases again, and the hardness also decreases.

### 3.3. Potentiodynamic Behavior

Potentiodynamic polarization curves are used to assess the electrochemical behavior of the coatings submerged in a 3.5 wt.% NaCl solution. The accompanying corrosion potential ($E_{corr}$), corrosion current density ($I_{corr}$), and polarization resistance ($R_p$) values are reported in Table 4, along with the potentiodynamic polarization plots displayed in Figure 9. It is obvious that in a 3.5 wt.% NaCl medium, the Fe-based composite coatings have displayed higher $E_{corr}$ and lower $I_{corr}$ as compared to Fe-based coating. Generally, the chemical makeup and organizational structure of materials have an impact on the corrosion potential and current density values. High $E_{corr}$ and low $I_{corr}$ suggest that the materials have lower corrosion tendencies, corrosion rates, and higher chemical stability [38,39]. Thus, it is reasonable to believe that Fe-based composite coatings have greater corrosion resistance.

**Table 4.** Corrosion potential, polarization resistance, and polarization current density of the coatings.

| Coating | 0 | 3% | 6% | 9% | 12% |
|---|---|---|---|---|---|
| Current density (A.cm$^{-2}$) | $6.09 \times 10^{-6}$ | $4.38 \times 10^{-6}$ | $3.03 \times 10^{-6}$ | $3.96 \times 10^{-6}$ | $5.04 \times 10^{-6}$ |
| Potential (V) | $-0.4656$ | $-0.4394$ | $-0.3908$ | $-0.4329$ | $-0.4242$ |
| Rp (ohms/cm$^2$) | 4283.4 | 5962.5 | 8613.1 | 6588.4 | 5172.6 |

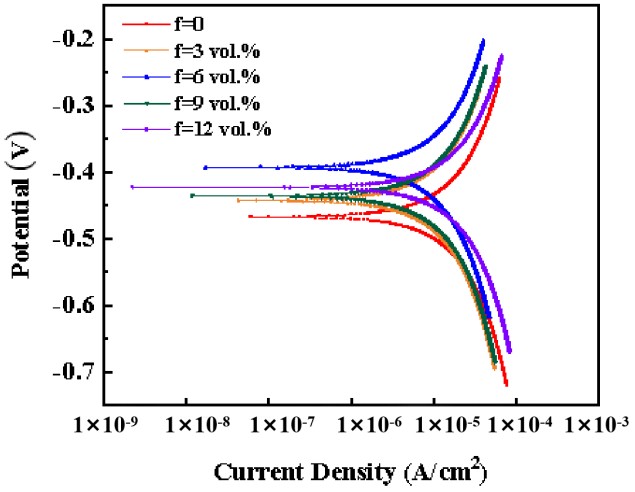

**Figure 9.** Potentiodynamic polarisation curves of the coatings in a 3.5 wt.% NaCl solution.

The Fe-based composite coatings have a substantially lower corrosion current density than the Fe-based coating, indicating a slower rate of corrosion. $E_{corr}$ is a thermo-dynamics parameter that reflects the property of corrosion resistance. According to Figure 9, the corrosion potential values of the Fe-based coating are $-0.4254$ V, while 6% $\gamma$-Fe coating and 9% $\gamma$-Fe coating are $-0.3363$ and $-0.4043$ V, respectively. This result indicates that Fe-based coatings are more prone to corrosion than Fe-based composite coatings. The resistance of the system to the corrosion process is characterized by polarization resistance; the polarization resistance is greater, and the coating has better corrosion resistance. The results from Table 4 show Fe-based coating has the lowest polarization resistance, and 6% $\gamma$-Fe coating and 9% $\gamma$-Fe coating have very similar $R_p$ values. When compared to Fe-based coating, Fe-based composite coatings have superior corrosion resistance. The opposite trend was observed when the additional amount of austenite particles was increased to 12%. This indicates that only the addition of austenite powder with the appropriate content can improve corrosion resistance.

The porosity of the coating is thought to play a significant role in the increased corrosion resistance of Fe-based composite coatings. The response area of coatings exposed to the corrosion solution decreases as the number of pores on the coated surface decreases. When

the potential and time grow throughout the electrochemical test, the corrosion medium gradually seeps into the interior pores of the coatings. As a result, the current density on the polarization curve gradually increases. This conclusion is somewhat supported with a comparison of the kinetic potential polarization curves in Figure 4.

### 3.4. Immersion Corrosion Experiment

The corrosion morphology of the samples is shown in Figure 10 after 1 and 7 days of immersion in a 3.5 wt.% NaCl solution. All samples have corrosion products visible on their surfaces, particularly the Fe-based coating, and it is clear that these corrosion products have grown holes and cracks. As the austenite particle content rises in the Fe-based composite coating, the corrosion surface becomes more and more flat and smooth. The inset of Figure 10 shows the macroscopic surface of the coating after 1 and 7 days of immersion corrosion, respectively. It reinforces the conclusion that the addition of austenitic powder has a significant effect on the corrosion resistance of the Fe-based coating.

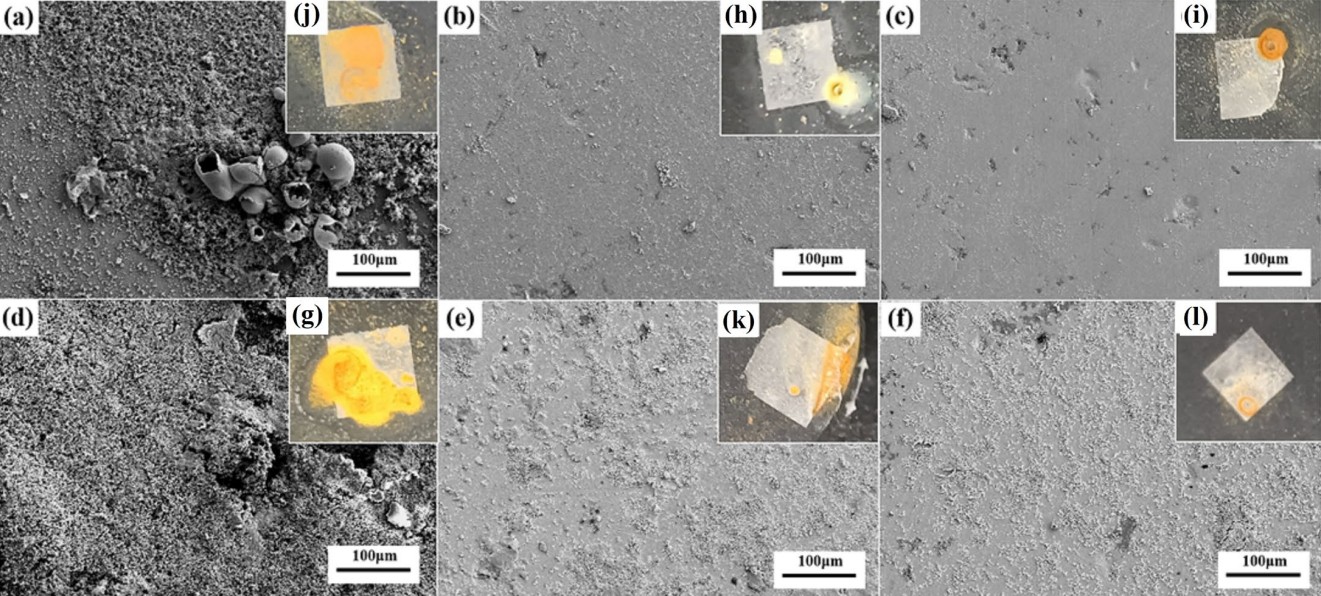

**Figure 10.** Morphologies of coatings immersed for 1 (**a**–**c**,**g**–**i**) and 7 (**d**–**f**,**h**,**j**,**l**) days: (**a**,**d**,**g**,**j**) Fe-based coating, (**b**,**e**,**h**,**k**) 6% γ-Fe coating, and (**c**,**f**,**i**,**l**) 9% γ-Fe coating.

More importantly, the corrosion product films of the Fe-based coating (Figure 11a,d) appear to be more porous and cracked compared to the Fe-based composite coatings. It confirms that the corrosion of the coating is more severe without the addition of austenite particles, which is consistent with the electrochemical measurements.

On the Fe-based coating surfaces, using the EDS test, the corrosion products are determined to be mainly composed of iron and oxygen elements (Figure 11d–f), indicating that the corrosion products formed were probably Fe oxides. The elemental content of Fe in oxidation products did not change significantly with the increase in austenite particle content. While the elemental content of Cr increased, it shows that Cr elements are crucial for corrosion resistance. The Si created the oxides during the immersion corrosion process, significantly enhancing the corrosion resistance of the coatings.

The inhomogeneity of the chemical makeup of the area around flaws in Fe-based coating is considered to be another important factor affecting their corrosion resistance in addition to porosity. Chromium is the most critical element for the corrosion resistance of Fe-based alloys. The readily generated chromium oxide is a stable and dense passivation film [40]. The creation of Cr-poor zones in Fe-based alloy coatings reduces corrosion resistance [41,42], thus making it more susceptible to localized corrosion, and the coating's overall level of corrosion resistance is dependent upon its weak areas. Consequently, remov-

ing these weak areas can improve the coating's corrosion resistance. Table 3 and Figure 7 show that the Cr content is higher and more evenly distributed in the Fe-based composite coatings. This proves that the austenitic composite results in a slight improvement in Cr depletion.

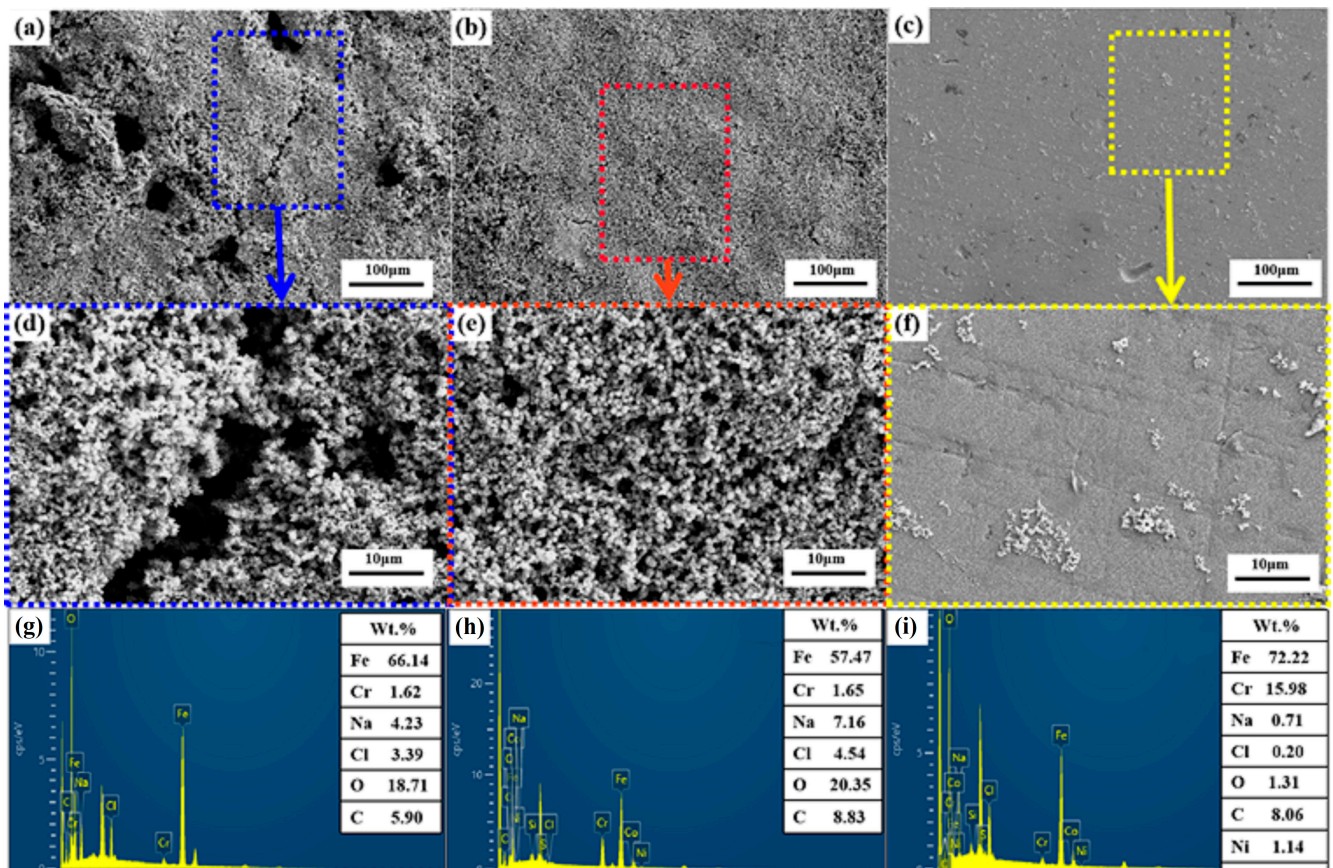

**Figure 11.** Morphologies of coatings immersed for three days and EDS scanning analysis: (**a,d,g**) Fe-based coating, (**b,e,h**) 6% γ-Fe coating, and (**c,f,i**) 9% γ-Fe coating.

*3.5. Erosion–Corrosion*

In the next work, the effects of austenite content on the mass loss rate of Fe-based coating are studied (Figure 12). As can be observed, when the austenite particle content is less than 9%, the erosion rate reduces as the austenite particle content increases. The mass loss rates of 6% γ-Fe coating and 9% γ-Fe coating in 3.5 wt.% NaCl solutions are 0.17 and 0.12 mg·mm$^{-2}$·h$^{-1}$, respectively, after eroding for 1 h. This shows that the Fe-based composite coatings have better slurry erosion–corrosion resistance compared to the Fe-based coating (0.27 mg·mm$^{-2}$·h$^{-1}$). It is very intriguing that the erosion mass loss of the coatings exhibits exactly the opposite tendency to the microhardness since the hardness has a significant impact on the slurry erosion resistance [20,43,44]. Consequently, it can be summarized that the hardness has a great impact on the erosion–corrosion resistance of the coatings. The report by Ricardo N. Carvalho et al. [45] indicated that the high alloying content of Cr, Ni, and Mo is a benefit in improving the corrosion resistance of alloys. In this work, there are more Cr and Si elements in the coating with 9% austenite particles, which suggests that the 9% γ-Fe coating has better erosion resistance than the Fe-based coating (Table 3 and Figure 7).

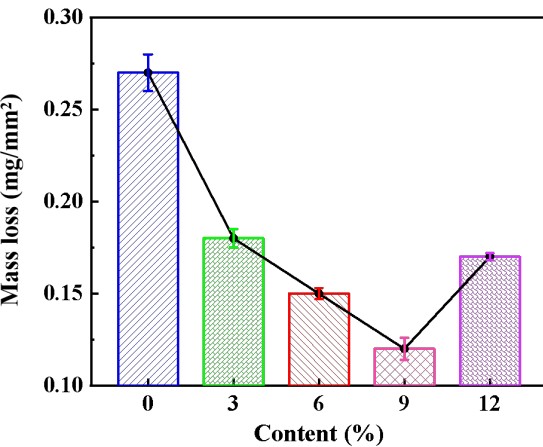

**Figure 12.** Mass loss curve of coating in a 3.5 wt.% NaCl solution.

SEM was performed to obtain more about the processes of slurry erosion–corrosion by analyzing erosion–corrosion surfaces of the coatings after they were subjected to erosion–corrosion tests for 1 h in 3.5 wt.% NaCl solutions. The SEM pictures of the coatings are shown in Figure 13. It can be observed clearly that the worn surface of Fe-based coating is relatively coarse. As compared to the Fe-based coating (Figure 13a,d), the 6% γ-Fe coating (Figure 13b,e) and 9% γ-Fe coating (Figure 13c,f) had considerably smoother eroded surfaces, which suggests that the Fe-based composite coatings have greater erosion–corrosion resistance. This agrees with the results of the mass loss. The wear surface of Fe-based coating has the morphology of deep plowing, accompanied by massive shedding and cracks. Fe-based composite coatings have shallow erosion pits; in contrast, the wear surface of Fe-based coating has deep plowing, which is accompanied by extensive shedding and fissures. After erosion, the surface of 9% γ-Fe coating is the smoothest.

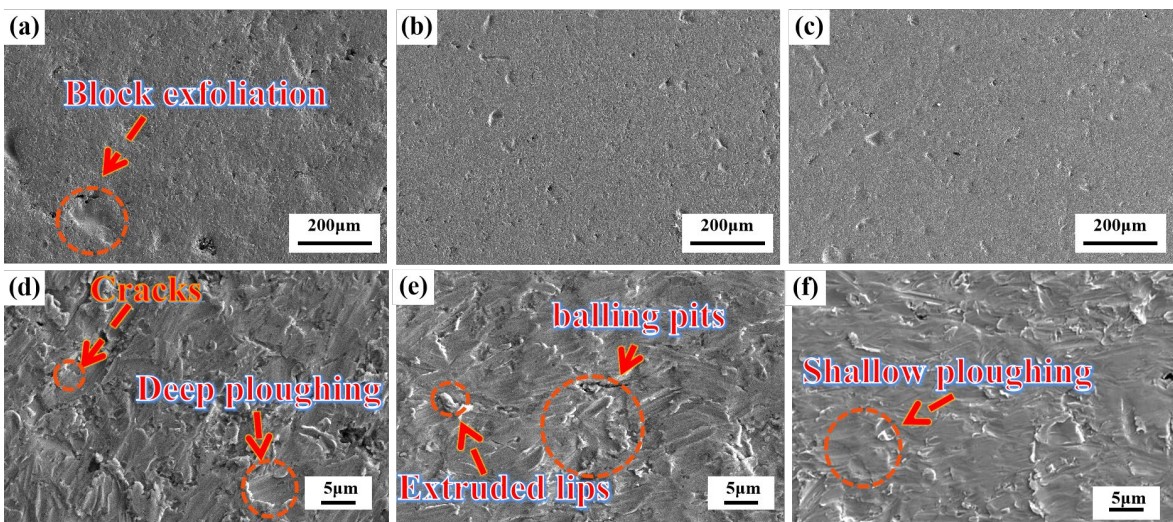

**Figure 13.** Micro-morphologies of the coatings after erosion at 60°: (**a**,**d**) Fe-based coating, (**b**,**e**) 6% γ-Fe coating, and (**c**,**f**) 9% γ-Fe coating.

It is evident that the erosion surface of the coating exhibits typical ductile erosion characteristics, which are characterized by mass shedding, raised plow lip formation, and micro-cutting, as shown in Figure 13. The coating has a few small fractures in it, as well as micro-cuts, fractures, and plow lips Figure 13d–f, which indicates a mixed pattern of ductile and brittle wear mechanisms. Moreover, Figure 13a depicts the degree of damage to the Fe-based coating under impingement; the Fe-based covering suffered the most serious damage. Compared to Fe-based coating, the severity of cracking is less severe in the worn

areas of composite coatings. A strong slurry impacts a weak surface, such as a surface with pores, and enlarges the pore size. At the same time, in Fe-based composite coatings, the high hardness improves erosion resistance, which is the reason why Fe-based composite coatings have less weight loss than Fe-based coating.

An atomic force microscope (AFM) was utilized to examine the coatings' roughness in order to better assess the erosion and corrosion of the coatings. Figure 14 exhibits the surfaces eroded at 3.5 wt.% sand-containing NaCl solution for 1 h. The square erosion surface under investigation had dimensions of $50 \times 50 \ \mu m^2$, and the color variation was used to indicate changes in the height of the erosion surface. Using AFM, the roughness parameters were acquired and are shown in Table 5 for analysis of the surface of the oxide film. Here, $R_q$ is the root-mean-square deviation of the assessed profile, and $R_a$ is its arithmetic mean deviation. The $R_a$ and $R_q$ values of the 9% γ-Fe coating were around 1.5 times as high as those of the Fe-based coating. The roughness parameter values of the Fe-based composite coating were at their lowest during the erosion, which suggests that the addition of austenite particles was helpful in enhancing the coating's resistance to erosion. The 9% γ-Fe coating had the lowest roughness rating and a comparatively smooth oxidized surface.

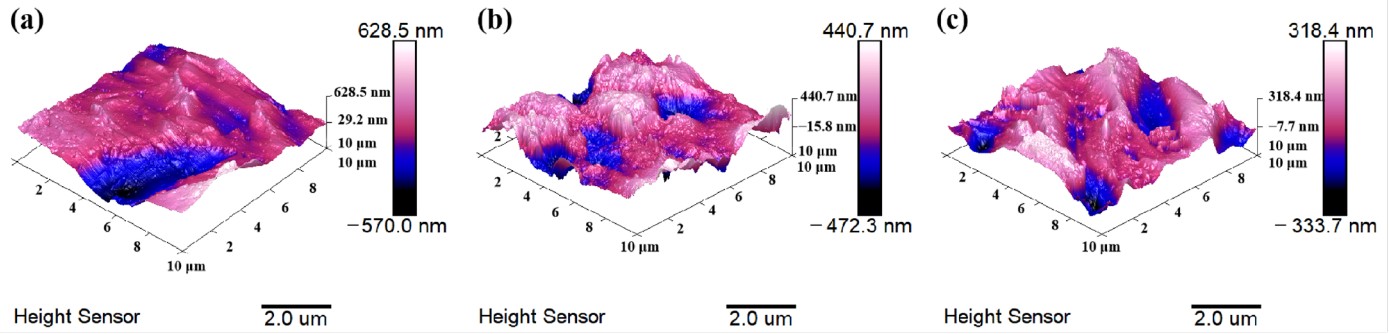

**Figure 14.** Atomic force micrographs taken on the erosion–corrosion surfaces of the coatings: (**a**) Fe-based coating, (**b**) 6% γ-Fe coating, and (**c**) 9% γ-Fe coating.

**Table 5.** Roughness parameters of the surface erosion film.

| Sample | $R_a/\mu m$ | $R_q/\mu m$ |
|:---:|:---:|:---:|
| f = 0 | 0.108 | 0.149 |
| f = 6 vol.% | 0.104 | 0.132 |
| f = 9 vol.% | 0.0714 | 0.0906 |

## 4. Conclusions

The supersonic plasma spraying (SPS) technique was used to combine austenite powder (0 vol.%, 3 vol.%, 6 vol.%, 9 vol.%, 12 vol.%) with an Fe-based self-fluxing alloy powder to create composite coatings with low porosity and excellent corrosion resistance. It is important to increase the service life of ship equipment in an environment of erosion and wear. This study looked into how austenite powder affected the microstructure and corrosion resistance of Fe-based coating. The following are the primary conclusions:

- The inclusion of austenite particles can lower the porosity and make the content of the hard phase (such as $Fe_3C$, $Cr_7C_3$, and $Cr_{23}C_6$) in the composite coatings gradually increase and become more uniformly distributed. The porosity of the coating decreased from 3.97% to 0.93%, and the hardness increased from $562HV_{0.1}$ to $636HV_{0.1}$.
- During the electrochemical corrosion experiments, it was found that Fe-based composite coatings had greater corrosion resistance than Fe-based coating in a 3.5 wt.% NaCl solution. The microstructure and chemical composition have a significant impact on how well coatings resist electrochemical corrosion. Fe-based composite coatings exhibit much superior corrosion resistance than Fe-based coatings applied in 3.5 wt.%

NaCl medium. At 9 vol.% and below, the corrosion resistance of Fe-based composite coating increases with the austenite powder content increase mainly because of the effective combination of low porosity, high content of corrosion resistance elements, and better chemical homogeneity.

- Erosion–corrosion tests indicate that Fe-based composite coatings have better erosion resistance than Fe-based coating. The Fe-based composite coating has a relatively smooth eroded surface when compared to the Fe-based coating, according to the results of scanning electron microscopy, and the results of atomic force microscopy further show that the Fe-based composite coating has a flatter surface morphology and a lower surface roughness value than the Fe-based coating after erosion.

**Author Contributions:** Conceptualization, S.L. and X.Z.; methodology, X.Z.; software, X.Z.; validation, J.W. and Z.Z.; formal analysis, X.Z.; investigation, H.C.; resources, T.L.; data curation, S.W.; writing—original draft preparation, X.Z.; writing—review and editing, X.Z.; supervision, K.Z.; project administration, K.Z.; funding acquisition, K.Z. All authors have read and agreed to the published version of the manuscript.

**Funding:** This research was funded by the GDAS' Project of Science and Technology Development (2022GDASZH-2022010103), The National Key Research and Development Program of China (2021YFB3701204).

**Institutional Review Board Statement:** Not applicable.

**Informed Consent Statement:** Not applicable.

**Data Availability Statement:** Not applicable.

**Conflicts of Interest:** The authors declare no conflict of interest.

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
