# Peer review of "Microstructure and Corrosion Behavior of Fe-Based Austenite-Containing Composite Coatings Using Supersonic Plasma Spraying"

_coatings, doi:10.3390/coatings13040694_

Round 1

Reviewer 1 Report

Overview:

In coatings-2300385 “Microstructure and corrosion behavior of Fe-based composite coatings by supersonic plasma spraying” manuscript, authors report that austenite addition into Fe60 alloy target improved the corrosion resistance and hardness of the coating. Authors present the explanation of the reported results based on γ-Fe melting. The study is coherent and interesting. However, its readability is currently interfered by significant concentration of the errors.

Comments:

1) Please consider mentioning austenite in the title, as it is the primary agent of the improvement of the studied coatings. I suggest the option of “…Fe-based austenite-containing composite coatings…”.

2) The coatings are named “Fe-based coating” for Fe material and “x% Fe-based composite coating” for austenite-containing coating. To me, this naming seems confusing, as it may imply there is x% Fe in the coating for the readers. I recommend either changing it (suggested option is “x% γ-Fe coating”), or introduce the names of the samples explicitly in the text. In the revised text, please clearly indicate that x% referred in the text is a percentage of the γ-Fe in the sprayed material, not in the resulting coating. (For example, the percentage seems misleading in line 182 “Fe-based composite coatings with fraction of 6 vol.% and 9 vol.%”.)

3)  Is “Austensite” (lines 11, 136, 238, 325, 327, 337, 345, 347) a proper term, or it should be replaced by “austenite” throughout the text?

4)    Lines 74-75, “Chemical composition of the Fe60 self-fluxing alloy powders is presented in Table 1”: how was is assessed? Either method or information source should be provided.

5)    Lines 76-77, “Using the three-dimensional mixing instrument equipment set the speed of 30r/min mixing time of 12h.”: please revise the sentence. Additionally, does “r/min” stand for “rotations per minute”?

6)    Lines 86-87: “Ar was employed as both the main gas and the carrier gas, and H2 was as the secondary gas.”: is it possible to indicate the residual and work pressure, if the deposition was carried out in vacuum chamber?

7)    In the Table 2, why is power being measured in kV units?

8) Line 105: “of size 10*10”: what are the measurement units?

9) Lines 106-107: please indicate how was a cross-section of the samples carried out.

10) Lines 139-140, “it is notable that the oxides are enriched in the linear phases from the results of EDS”: is “linear phase” a proper term? What does it mean?

11) The IPF abbreviation should be defined in Line 154, not in Line 175.

12) Lines 180-182: “The measurement of the 180 porosity was evaluated on the basis of the image analysis”: could you please describe the methodology of porosity assessment from SEM in Section 2?

13) In Fig. 3abc, “matrix” notation is not visible, please add a black outline. In Fig. 3b, the scale bar is corrupted by the dashed line.

14) In fig. 3a and associated text, what do you mean by “matrix”? Is it a substrate, or is a same kind of matrix that is mentioned in line 186-187 (“austenite particles used in this research are embedded into the Fe matrix very well”)? I also suggest revising the fragment “the bonding strength between the coating and the matrix” (line 148-149).

15) Line 162: was the “OM” abbreviation defined previously?

16) Lines 203-204, “When austenite was added to the coating, there appeared γ-Fe and Fe3C phases”, line 229, “some metallic carbides phase, such as Cr23C6, Cr7C3 and Fe3C are formed in the process of the thermal spraying”: as there is no carbon in austenite precursor, please indicate how the carbide phases were formed in more detail.

17) Table 3: f parameter was not introduced. Why don’t the values add up to 100% total? Are BCC/FCC related to Fe?

18) Lines 238-239 grain size strengthening: what do you mean by this term and what results indicate this process is taking place?

19) Line 231: is “dispersion strengthening” a proper term? What does it mean?

20) Line 247: what do you mean by “lap porosity”?

21) In Table 4, there is an unconventional notation for the sample names (“Coating 2”?).

22) Table 4 caption indicates “Corrosion potential and polarization current density of the coatings”. Please indicate what is the Rp value presented in the Table.

23)  In the caption of Fig. 10, please indicate what is shown in the inset. Are the insets of Fig. 10 discussed in the text?

24) Line 302, please revise the fragment “EDS was detected mainly by Fe and O elements”.

25) In Fig. 13, the red text is barely visible, could you please make more prominent white outline of the letters or add a semi-transparent background below the text?

26) In Fig. 14, the numbers on the XYZ scale bars are not visible, please increase the font size.

27) Line 393, “the largest decrease reached 76.6%” fragment seems redundant.

28) In the conclusions, it would be beneficial to indicate what potential applications of the studied composite coatings emerge from the obtained results.

29) In the Graphical Abstract, I recommend to get rid of the unreadable elements (scale bar tick labels), increase the scale bar titles whenever possible, indicate the role of austenite, remove  less important fragments and sub-figures with unreadable text (for example, the one originating from fig. 13).

Author Response

RESPONSE TO REVIEWERS’ COMMENTS

Dear Editors and Reviewers:

   Thank you for your letter and for your comments concerning our manuscript entitled “Microstructure and corrosion behavior of Fe-based composite coatings by supersonic plasma spraying” (coatings-2300385). Those comments are all valuable and very helpful for revising and improving our paper, as well as the important guiding significance to our researches. We have studied comments carefully and have made corrections which we hope meet with approval. Changes are highlighted with green background in the revised version. The main corrections in the paper and the responses to the reviewers’ are as following:

Reviewer #1:

Comment 1: "Please consider mentioning austenite in the title, as it is the primary agent of the improvement of the studied coatings. I suggest the option of “…Fe-based austenite-containing composite coatings…”"

Reply: I am grateful for your advice, we make appropriate modifications to the title: "Microstructure and corrosion behavior of Fe-based austenite-containing composite coatings by supersonic plasma spraying"

Comment2: "The coatings are named “Fe-based coating” for Fe material and “x% Fe-based composite coating” for austenite-containing coating. To me, this naming seems confusing, as it may imply there is x% Fe in the coating for the readers. I recommend either changing it (suggested option is “x% γ-Fe coating”), or introduce the names of the samples explicitly in the text. In the revised text, please clearly indicate that x% referred in the text is a percentage of the γ-Fe in the sprayed material, not in the resulting coating. (For example, the percentage seems misleading in line 182 “Fe-based austenite-containing composite coatings with fraction of 6 vol.% and 9 vol.%”.)"

Reply: Thank you very much for reminding us to take note of this issue! We have detailed the naming of the coating in the article on lines 95-98 and have made corresponding changes in the article in the corresponding places.

Comment3: Is “Austensite” (lines 11, 136, 238, 325, 327, 337, 345, 347) a proper term, or it should be replaced by “austenite” throughout the text?.

Reply: Thank you for your suggestions, We have modified all the place you mentioned accordingly by changing “Austensite” to “austenite”.

Comment4: Lines 74-75, “Chemical composition of the Fe60 self-fluxing alloy powders is presented in Table 1”: how was is assessed? Either method or information source should be provided.

Reply: Thanks for the reminder, The source of the chemical composition of the Fe60 powder and austenite powder is Nangong Xindun Alloy Welding Spray Co and Hebei Yangyou Metal Materials Co. And we have also added powder sources to the article.

Comment5: Lines 76-77, “Using the three-dimensional mixing instrument equipment set the speed of 30r/min mixing time of 12h.”: please revise the sentence. Additionally, does “r/min” stand for “rotations per minute”?

Reply: Thank you very much for your careful pointing out the error of the above sentence. We have modified the sentence you mentioned, the specific modifications are as follows:“Set the speed of the three-dimensional mixing instrument equipment to 30 rpm and the mixing time to 12 hours.” 

Comment6: Lines 86-87: “Ar was employed as both the main gas and the carrier gas, and H2 was as the secondary gas.”: is it possible to indicate the residual and work pressure, if the deposition was carried out in vacuum chamber?

Reply: The spraying process is not in a vacuum chamber. Supersonic plasma spraying uses a rigid non-transferable plasma arc as the heat source to heat the working gas to form a high temperature and high speed plasma jet, thus heating and accelerating the powder fed into the jet to form a molten particle stream that hits the surface of the substrate at high speed to form a coating thermal spraying technology.

Comment7: In the Table 2, why is power being measured in kV units?

Reply: I apologize for the error caused by our negligence. we have replaced kV with kW.

Comment8: Line 105: “of size 10*10”: what are the measurement units?

Reply: I apologize for our negligence.We have added units(μm) to the size 10*10.

Comment9: Lines 106-107: please indicate how was a cross-section of the samples carried out

Reply: In order to make the sentence unambiguous, we modified “the cross-section of the sample” to that of “the cross-section of the coatings”. As described in the section 2, the cross-section of coatings is observed by grinding and polishing the cross-section with the coating samples, and finally the morphology is observed as shown in Fig 2"

Comment10: Lines 139-140, “it is notable that the oxides are enriched in the linear phases from the results of EDS”: is “linear phase” a proper term? What does it mean?

Reply: Linear phase is not a proper term, We have modified it accordingly, the specific modifications are as follows:“It is noteworthy that from the EDS results, the oxide enrichment becomes line shapes”

Comment11: The IPF abbreviation should be defined in Line 154, not in Line 175.

Reply: Thanks for the careful reminder! We have modified the article accordingly. IPF abbreviations have been defined in line 154 and abbreviations are used in line 175.

Comment12: Lines 180-182: “The measurement of the 180 porosity was evaluated on the basis of the image analysis”: could you please describe the methodology of porosity assessment from SEM in Section 2?

Reply: Thanks for the suggestion! We have added the detailed method for measuring porosity in Section 2, the specific modifications are as follows: “By randomly selecting more than 10 cross-sectional SEM images of the coating at 800× magnification, the porosity results were calculated and averaged using porosity assessment software (Image J2)..”

Comment13: In Fig. 3abc, “matrix” notation is not visible, please add a black outline. In Fig. 3b, the scale bar is corrupted by the dashed line.

Reply: Thanks for the careful reminder! We have modified the figure accordingly in the article. The specific modifications are as follows fig. 1:

Fig .1

Comment14: In fig. 3a and associated text, what do you mean by “matrix”? Is it a substrate, or is a same kind of matrix that is mentioned in line 186-187 (“austenite particles used in this research are embedded into the Fe matrix very well”)? I also suggest revising the fragment “the bonding strength between the coating and the matrix” (line 148-149).

Reply: Thanks for the suggestion! In this article both matrix and substrate stand for the coating substrate 45 steel. To avoid confusion, we have replaced all matrix with substrate throughout the article. Moreover, this sentence “the bonding strength between the coating and the matrix” has been modified to “the binding capacity of substrate and coating” 

Comment15: Line 162: was the “OM” abbreviation defined previously?

Reply: Thank you for your careful reminder! The “OM” abbreviation is not defined and we have replaced it with the more specific software name(OIM Analysis6).

Comment16: Lines 203-204, “When austenite was added to the coating, there appeared γ-Fe and Fe3C phases”, line 229, “some metallic carbides phase, such as Cr23C6, Cr7C3 and Fe3C are formed in the process of the thermal spraying”: as there is no carbon in austenite precursor, please indicate how the carbide phases were formed in more detail.

Reply: Carbon from Fe-based alloy powder. During the spraying process, the carbon in the Fe-based alloy reacts with the fe and cr to produce carbide.

Comment17: Table 3: f parameter was not introduced. Why don’t the values add up to 100% total? Are BCC/FCC related to Fe?

Reply: Thank you very much for reminding us to take note of this issue! We have introduced the meaning of f in the article. Due to unavoidable defects of the coatings and restrict of equipment, it is difficult to achieve a 100% calibration rate for each sample, and therefore the percentage of phases in the coating does not reach 100%. But our calibration results are tested in the same environment, so the results are reliable. And BCC/FCC stands for α-Fe/γ-Fe, respectively.

Comment18: Lines 238-239 grain size strengthening: what do you mean by this term and what results indicate this process is taking place?

Reply:Thanks for the reminder! What we want to express is the fine grain strengthening, which has been described more precisely in the article. And Fig. 3 can strongly support this conclusion.

Comment19: Line 231: is “dispersion strengthening” a proper term? What does it mean?

Reply: Dispersion strengthening is a proper term. Dispersion strengthening is a means of strengthening a material by adding hard particles to a homogeneous material. It is a metallic material strengthened with an ultra-fine second phase (reinforced phase) that is insoluble in the base metal. In order to make the second phase uniformly distributed in the base metal, it is usually manufactured by powder metallurgical methods. The second phase is generally a high melting point oxide or carbide or nitride, and its strengthening effect can be maintained to a higher temperature. Dispersion strengthening is a method of strengthening alloys with a large strengthening effect and is very promising. If the compound in the solid solution grains are diffuse mass or grain distribution, it can significantly improve the strength and hardness of the alloy, but also plasticity and toughness decline is not large, and the finer the particles, the more diffuse and uniform distribution, the better the strengthening effect.

Comment20: Line 247: what do you mean by “lap porosity”?

Reply: Thank you for your care in pointing out the inappropriate descriptions about “lap porosity” in this article,in order to make this article more rigorous in the description, we have modified “lap porosity” to “porosity” in the article.

Comment21: In Table 4, there is an unconventional notation for the sample names (“Coating 2”?).

Reply: I apologize for the error caused by our negligence. It should originally be represented by “coating”. and we have made corresponding changes in Table 4.

Comment22: Table 4 caption indicates “Corrosion potential and polarization current density of the coatings”. Please indicate what is the Rp value presented in the Table.

Reply: I apologize for our negligence. We have made corresponding changes in Table 4. The caption of Table 4 has been refined to “Corrosion potential, polarization resistance and polarization current density of the coatings”.

Comment23: In the caption of Fig. 10, please indicate what is shown in the inset. Are the insets of Fig. 10 discussed in the text?

Reply: Thanks for the careful reminder! We have described what is shown in the inset in the caption of Fig. 10. and added a discussion of the inset to the text. The specific modifications are as follows: “The inset of Fig. 10 shows the macroscopic surface of the coating after 1 and 7 days of immersion corrosion.respectively. It reinforces the conclusion that the addition of austenitic powder has a significant effect on the corrosion resistance of the Fe-based coating.” 

Comment24: Line 302, please revise the fragment “EDS was detected mainly by Fe and O elements”

Reply:I am grateful for your advice, we make appropriate modifications to this fragment in the article :“By EDS test, the corrosion products are mainly composed of iron and oxygen element.”

Comment25: In Fig. 13, the red text is barely visible, could you please make more prominent white outline of the letters or add a semi-transparent background below the text?

Reply:Thanks for the suggestion! We have modified Fig. 13 accordingly in the article. As showed in fig. 2.

.

fig. 2

Comment26: In Fig. 14, the numbers on the XYZ scale bars are not visible, please increase the font size.

Reply:Thanks for the reminder! We have modified Fig. 14 accordingly in the article. As showed in fig. 3.

Fig. 3

Comment27: Line 393, “the largest decrease reached 76.6%” fragment seems redundant.

Reply:Thanks for your suggestion! We have delected this fragment in the article.

Comment28: In the conclusions, it would be beneficial to indicate what potential applications of the studied composite coatings emerge from the obtained results.

Reply: Thanks for the reminder! We have added the corresponding content about the potential applications in the article. The specific modifications are as follows: “It is Significant that in extending the service life of ship equipment under the dual environment of wear and erosion.” 

Comment29: In the Graphical Abstract, I recommend to get rid of the unreadable elements (scale bar tick labels), increase the scale bar titles whenever possible, indicate the role of austenite, remove less important fragments and sub-figures with unreadable text (for example, the one originating from fig. 13).

Reply: Thank you for your constructive suggestions! We have modified the Graphical Abstract accordingly. As showed in fig. 3.

fig. 3

Special thanks to you for your comments which are helpful in correcting mistakes in our article.

Reviewer 2 Report

The manuscript is well organized, the experiment is logical, the successive studies are logically arranged. Graphically, the work also meets the standards of the journal. The authors indicate a gap in knowledge and propose to fill it with their research. The experiment is well described and reproducible (please only the name of the XRD diffractometer used).

I have a question about XRD. Are the Authors completely sure about the presence of FCC-structured Fe? The diffraction meter was equipped with a CuKa lamp, is the peak marked Fe corrected? EBSD studies indicate a very small presence of this phase, if it was actually around 0.19 - 0.31% XRD probably wouldn't even detect it.

I have no comments to describe the structure and properties of the coatings. I believe that the article can be published with minor changes.

Author Response

Reviewer #3:

Comment: The author has explained clearly explained importance of Microstructure and corrosion behavior of Fe-based composite 2 coatings by supersonic plasma spraying.

It was observed some grammatical mistakes and incomplete sentences in the results and discussion part. Manuscripts should be throughly checked for this typo error before publication. Navality of the work to be explored in an understandable manner in the introduction and abstract part.

Reply: We have tried our best to revise the whole manuscript carefully to avoid language errors on the use of tense, article, singular & plural and others. In addition, we have asked skilled authors of English language papers to check the English.

Reviewer 3 Report

The author has explained clearly explained importance of Microstructure and corrosion behavior of Fe-based composite 2 coatings by supersonic plasma spraying.

It was observed some grammatical mistakes and incomplete sentences in the results and discussion part. 

Manuscripts should be throughly checked for this typo error before publication

Navality of the work to be explored in an understandable manner in the introduction and abstract part.

Author Response

RESPONSE TO REVIEWERS’ COMMENTS

Dear Reviewers:

   Thank you for your letter and for your comments concerning our manuscript entitled “Microstructure and corrosion behavior of Fe-based composite coatings by supersonic plasma spraying” (coatings-2300385). Those comments are all valuable and very helpful for revising and improving our paper, as well as the important guiding significance to our researches. We have studied comments carefully and have made corrections which we hope meet with approval. Changes are highlighted with green background in the revised version. The main corrections in the paper and the responses to the reviewers’ are as following:

Reply: We have tried our best to revise the whole manuscript carefully to avoid language errors on the use of tense, article, singular & plural and others. In addition, we have asked skilled authors of English language papers to check the English.

Special thanks to you for your comments which are helpful in correcting mistakes in our article.

Reviewer 4 Report

1. Although the introduction contains important information, I believe that the review of the research to date is too poor, you should improve it
2. Line 77 reads [r/min], it should be RPM
3. In lines 79-81 it is written that the surface was first cleared with alcohol and then sandblasted. Shouldn't it be the other way around? Why was it degreased before sandblasting?
4. On line 85, "compressed" is written with a capital letter
5. The description of the spraying method is too brief, please elaborate
6. During the corrosion tests, the sample was cleaned with alcohol (line 126), is this a common method? What concentration of alcohol was it? The water contained in it can cause corrosion, have the authors estimated the impact of this factor on the test results?
7. The results of the research are very extensively described and do not raise any major objections.

Reviewer 5 Report

The article needs to be thoroughly improved. Here are just a few of my comments.

Line 193-197: Does the amount of austenite change its melting point? Does the amount of it simply affect the number of filling cracks and holes? This needs to be explained somehow!!!

 Fig.6. What is the difference in the diffraction spectra in Fig.6. What conclusions can be drawn apart from the fact that there are certain phases listed in this figure?

Line 222-233: should be corrected, e.g. the sentence "In order to analyze the effects of the addition of austenite on the Fe-based coating, microhardness for the coatings were studied (Fig.8)”.

My proposal: The influence of austenite on the microhardness of the obtained coating was investigated (Fig.8).

Line 234: doped with 9 vol.%

Line  235: … be attributed mainly to the following:

 -admixture 9 vol.% of austenite powder not only leads to the precipitation of Cr23C6, Cr7C3 and 236

Fe3C phases, but increasing the relative contents of compounds in the Fe-based coating, causing second phase strengthening,  (what is the second phase?)

- austensite powder refines the microstructure of the Fe-based coating, resulting in grain size strengthening.

Define the term “grain size strengthening”

Line 240-248: This needs to be explained more clearly and precisely

Line 258: the chemical components and organizational structure of materials - what did the authors mean?                    

General note:

 the rest of the article is written in a similar style. I am not able to show point by point inaccuracies and imprecision. It should be written more succinctly and not in the style of more or less etc. The article needs proofreading by a native speaker.

Round 2

Reviewer 1 Report

Authors of coatings-2300385Microstructure and corrosion behavior of Fe-based austenite-containing composite coatings by supersonic plasma spraying” have substantially improved the manuscript and carefully answered most of my questions. Several issues should still be corrected.

1) In the reply to my comment 16, authors indicated that carbon content in the resulting coatings originates from the Fe60 alloy powder. However, the inset of Fig.1 (EDX results) indicates that there is no carbon in the powder. Authors should discuss this effect in more detail. I assume that carbon originates from the impurities present on the surfaces of the materials, and these impurities are somehow rearranged into the crystalline structure under the plasma beam.

2) Line 114: “Every coating sample was measured at least 8 and then to be averaged.”: please revise the sentence.

3) Line 151, the phrase “the oxide enrichment becomes line shapes” is unclear.

4) Line 253: what is “lemalla oxide”?

5) In the caption or in the X axis title of Fig. 8, indicate which content changes.

6) Line 407-408, “It is Significant that in extending the service life of ship equipment under the dual environment of wear and erosion.”: please rephrase.

7) In my opinion, small unreadable elements, such as titles and numbers of/on the axes, should be removed from the Graphical Abstract, and every notation that is crucial for the understanding and for the promotion of the paper should be written with the enlarged font.

Author Response

请参阅附件

Reviewer 5 Report

The article looks much better. Thank you for the changes made

Author Response

RESPONSE TO REVIEWERS’ COMMENTS

Dear Reviewer:

   Thank you for your letter and for your comments concerning our manuscript entitled “Microstructure and corrosion behavior of Fe-based composite coatings by supersonic plasma spraying” (coatings-2300385).

Special thanks to you for your comments which are helpful in correcting mistakes in our article.